# Detection of Microrelief Objects to Impede the Movement of Vehicles in Terrain

**Filip Dohnal**[ID]**, Martin Hubacek \***[ID] **and Katerina Simkova**

Faculty of Military Technology, University of Defence, Kounicova 65, 662 10 Brno, Czech Republic;
filip.dohnal@unob.cz (F.D.); kat.sim@email.cz (K.S.)
**\*** Correspondence: martin.hubacek@unob.cz

**Abstract:** Relief of terrain as a part of the landscape greatly affects the possibilities of vehicles moving off the road. The main influence on the movement is the slope of terrain and the occurrence of microrelief objects. While the slope limits can be easily modeled in the GIS environment, it is difficult to express the effect of the microrelief on the possibilities of moving vehicles. The aim of this work was to find procedures for identification of impassable microrelief objects using GIS tools and precise digital elevation models. Technical parameters defining the ability of a vehicle to overcome microrelief objects are known and these are mainly defined by the dimensions of the vehicle such as a wheel base, a ground clearance, approach angle, and others. Large-scale digital elevation models have not been able to reliably express the location and shape of microrelief objects until recently. Their accuracy of height in nodes achieved meter or decimeter values. The change occurred with the use of airborne laser scanning technology for digital elevation model creation. The accuracy of models created using this technology achieves centimeter values. These can be used for detection of microrelief objects. One of these models is the DMR5 from the territory of the Czech Republic. Its declared total mean height error is 0.18 meters. This model, together with the GIS tools and the technical parameters of individual vehicles, was used to search for such microrelief objects that act as a barrier to movement. Procedures for detecting impassable microrelief objects were created by ArcGIS tools. Modeling tools and mathematical methods were used to create procedures for detection of microrelief objects. These have been applied to selected locations in the Czech Republic. Raster layers representing individual impassable microrelief objects are the result of modeling. The modeling results were verified in the terrain using military vehicles. Field tests confirmed the high reliability of the proposed procedure. Therefore, the calculation process was optimized and will be introduced in the future as one of the input calculations of the complex model of passability in the Army of the Czech Republic.

**Keywords:** microrelief; passability; digital elevation model; vehicle; military spatial analysis

## 1. Introduction

Relief is one of the basic components of the landscape [1,2]. By its fragmentation, the landscape is shaped and to a great extent conditioned by the appearance of other landscape elements in the area. Its configuration affects many human activities. The issue of transporting people and materials is one of the activities that the relief affects. Its influence is enhanced for off-road transports. This way of transport is particularly characteristic for military units and rescue troops during crisis tasks. Good knowledge of maneuver options is thus a prerequisite for a successful accomplishment of assigned tasks. This is why military geographers in particular have been working for decades on a variety of terrain impact assessments for the mobility of military vehicles. In the beginning they were mainly word comments and thematic maps (Figure 1) [3,4], at present they are mainly outputs of complex models of terrain passability (Figure 2) [5–9]. All known models of passability [5,6,9–12] assess the impact of the terrain

on the movement of vehicles in a comprehensive way. The input layers for modeling are slope, soil, vegetation, hydrology, and more. However, none of the models consider the objects of microrelief. Outside of the military domain, the issue of vehicle movement over terrain is a subject of research in agriculture and forestry. Research in this area is mainly focused on the impact of soil on the movement of vehicles and on agricultural and forestry activities [13,14]. Individual studies deal with tire and chassis characteristics, wheel slip, slope, soils, and issues of terramechanics in general.

Concerning the impact of the relief on mobility, mainly the slope and the influence of the microrelief elements are evaluated. In general, the microrelief objects can be defined as objects of relief, which height differs from the vicinity in not more than a few meters. These objects are predominantly linear or point-like and occur mainly in the form of streams, erosion, terraces, excavations, embankments, pits, and other objects. From the point of view of the movement in the terrain, the shape, size, and direction of microrelief objects influence the maneuvers of military units as well as of individual vehicles. While the slope effect can be easily modeled using GIS tools, it is relatively difficult to do so for a microrelief. The technical parameters of vehicles define type and geometry of chassis and they are necessary to model the ability to overcome a particular microrelief object. These are known for each vehicle. However, the use of digital elevation models (DEM) covering large areas is problematic. Most of these DEMs covering the territories of states, continents, or the whole world were created using methods of cartometry, radar interferometry, or stereophotogrammetry. DEMs are usually in the form of a grid of points with a spacing of meters up to hundreds of meters and their accuracy of height in points achieved meter or decimeter values. No interpolation techniques can identify the terrain edges and microrelief objects from these models. The change was made using airborne laser scanning technology to create digital elevation models. This technology can quickly and efficiently collect precise data about the location and the height of the relief and the objects on it.

In the years 2010–2013, this technology was used for the first nationwide scanning of the territory of the Czech Republic. The result of the project involving the Czech Office for Surveying and Cadastre, the Ministry of Defense, and the Ministry of Agriculture is the new generation of digital elevation models, DMR4 and DMR5, completed in 2016 [15]. DMR5 is created in the form of irregularly spaced points with a total mean height error in the exposed terrain of 0.18 m and 0.3 m in the forested area [16]. The tests made by the creators as well as the independent verifications confirmed the high accuracy of these models [17,18]. These models, in particular DMR5, alter the modeling and field analysis capabilities in many areas. They also can be used in areas where until now it was necessary to perform detailed geodetic measurements before commencing analytical work [19].

The aim of this work is to find procedures for identification of impassable microrelief objects using GIS tools and precise digital elevation models.

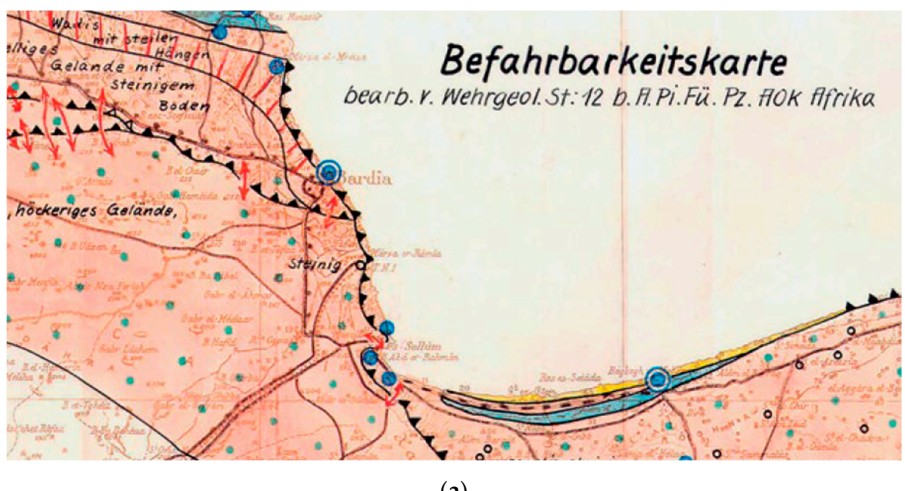

(a)

**Figure 1.** *Cont.*

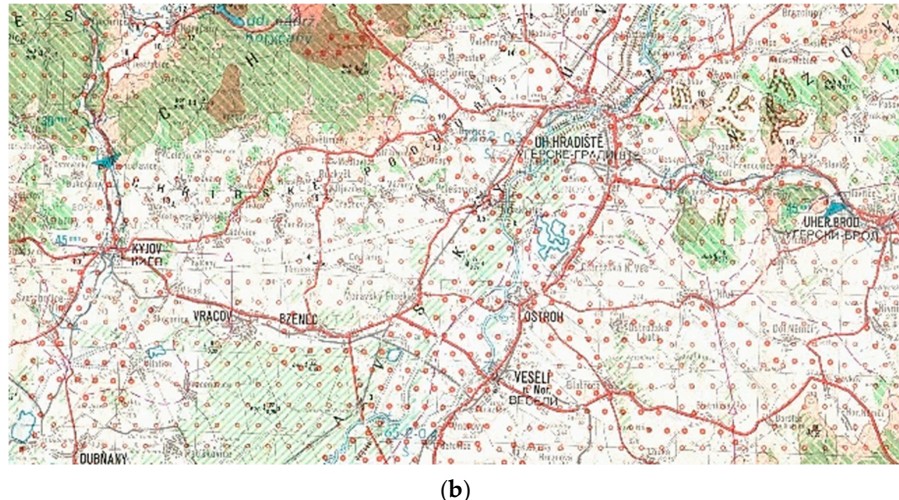

(**b**)

**Figure 1.** Examples of thematic maps of passability: (**a**) Section of a hand-colored German going-map (Befahrbarkeitskarte) 1:500,000 (World War II) [3]; (**b**) Czechoslovak map of terrain passability 1:200,000 (1980s), Archive of the Department of Military Geography and Meteorology.

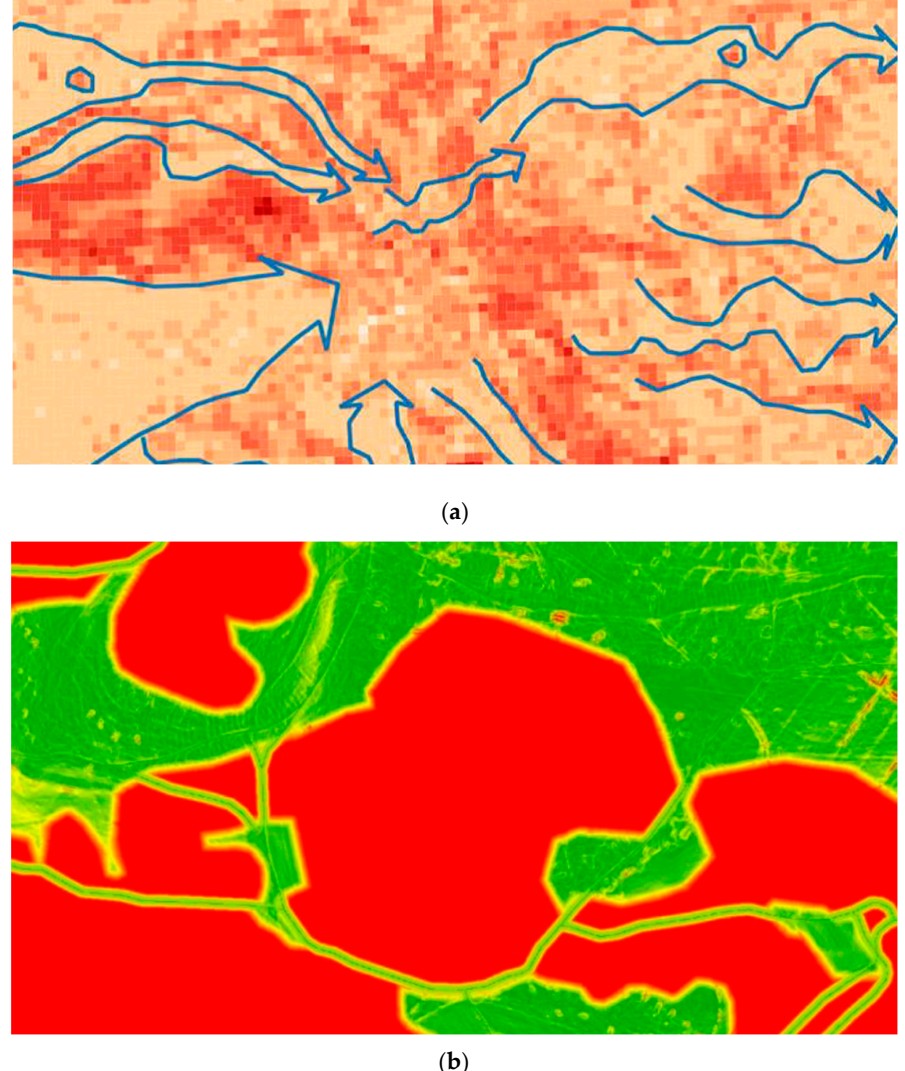

(**a**)

(**b**)

**Figure 2.** Examples of modeling of passability in GIS: (**a**) Map output of the Polish model of passability [6]; (**b**) Map output of the Czech model of passability [8].

## 2. Effect of Microrelief Objects on Movements

Microrelief objects can be overcome in two different ways [20]. The first is the direct overtaking of the object by the vehicle. The second way is to bypass the object. In both cases, the progress of the units slows down, and in the latter case the need for maneuver even grows. To assess the vehicle's ability to overcome microrelief objects, it is necessary to know the tactical and technical parameters of the vehicle, dimensions, and shapes of the microrelief objects. Based on this knowledge, it is possible to decide whether vehicles are able to overcome objects of microrelief or they need to bypass them. The basic technical parameters of the vehicles to determine the ability to overcome microrelief objects are [21]:

- chassis type (wheeled, tracked);
- number of axles;
- wheelbase;
- angles of approach;
- climbing ability—vertical step;
- ground clearance of the vehicle

These values are generally known and precisely quantified for each vehicle. With a certain schematization and simplification of the problem, in the situation when it is impossible to overcome the microrelief object for wheeled vehicles, the following examples may occur:

- the perpendicular edge is higher than the vehicle climbing ability (Figure 3a);
- a deep notch is wider than the vehicle's crossing ability—trench width (Figure 3b);
- the angle between the relief and the microrelief is greater than the approach angle of the vehicle (Figure 3c);
- the angle of the top edge of the microrelief object is less than twice the angle between the center of the wheelbase and the lower edge of the tire on the axle (Figure 3d)

In case of tracked vehicles, the situation is similar, except for the last case that does not occur in this type of chassis.

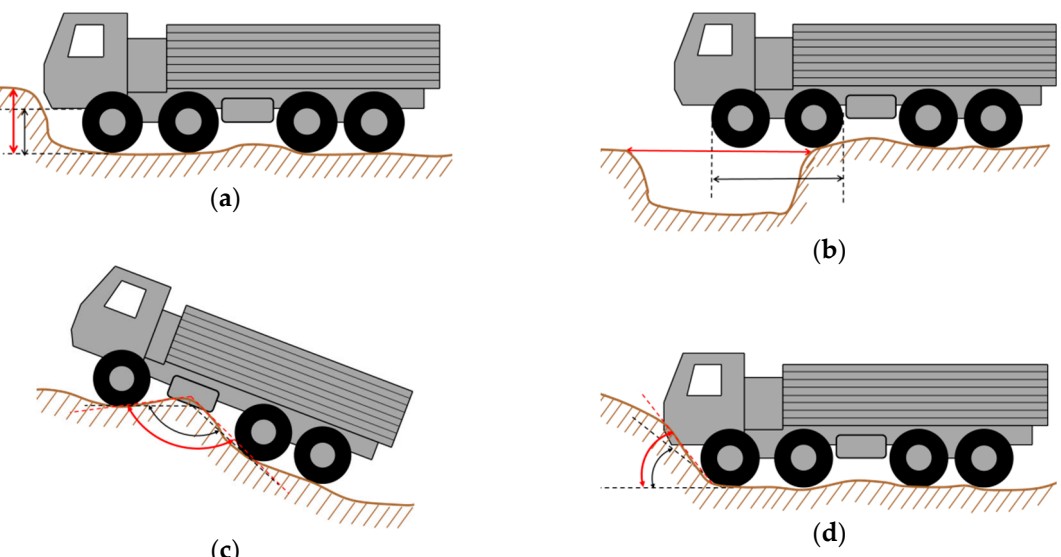

(a)

(b)

(c)

(d)

**Figure 3.** Visualization of vehicle limits to overcome microrelief objects (black arrow): (**a**) The perpendicular edge is higher than the vehicle climbing ability; (**b**) A deep notch is wider than the vehicle's crossing ability; (**c**) The angle between the relief and the microrelief is greater than the approach angle of the vehicle; (**d**) the angle of the top edge of the microrelief object is less than twice the angle between the center of the wheelbase and the lower edge of the tire on the axle.

### 3. Identification of Impassable Microrelief Objects

Within the proposed and tested solution, three of the four cases have been resolved so far. The suggested way of identifying microrelief objects uses DMR5 data. From this model, a raster model with a pixel size of 0.5 meters (0.4 m or 0.6 m) is calculated. This value was chosen based on the detected DMR5 error in the area of microrelief shapes [16] and technical parameters of the vehicle. The considered cases are:

- high perpendicular field slope (Figure 3a);
- large approach angle (Figure 3c);
- sharp field edge (Figure 3d)

The issue of the width of the recessed microrelief objects will be solved in the future. The entire calculation process is implemented in the ArcGIS programming environment and uses standard features.

*3.1. Identification of Impassable Perpendicular Stages*

The identification of impassable perpendicular stages was the easiest task to solve. For the calculation, it is necessary to know only the vehicle climbing ability parameter ($C$). The value of this parameter is compared with the raster values obtained after calculating the height difference of neighboring pixels ($\Delta H$) in the DMR5 raster with a pixel size of 0.5 m. The difference in altitude values is calculated by the FocalStatistic function within the $3 \times 3$-pixel search window. Based on the knowledge of the DMR5 accuracy and vehicle field tests, a possible error corresponding to 20% of the output value has to be included in the calculation.

The calculated values are then classified as follows:

- impassable microrelief object ($\Delta H \geq C + 0.2C$);
- probably impassable microrelief objects ($\Delta H < C + 0.2C \land \Delta H > C - 0.2C$);
- passable microrelief objects ($\Delta H < C - 0.2C$)

The amount of these objects in the field is quite small compared to the following cases. Most of these objects are artificial and occur mainly in settlements and other intensely used areas.

*3.2. Identification of Impassable Sharp Edges*

For the detection of terrain edges where the wheeled vehicle can be trapped by a chassis, a procedure based on the limit values of two technical parameters of wheeled vehicles (ground clearance vehicle and wheelbase) was proposed. No deformation of springs, tires, and terrain surface due to vehicle load is considered in the calculation [22]. The principle of the calculation is based on the comparison of terrain altitude in the place of the terrain edge with the altitude of the chassis at the hypothetical point of contact with the terrain between the axles (Figure 4). The calculation algorithm is based on the use of map algebra and DMR5 in a raster format. The value of the pixel size is chosen depending on the length of the wheelbase of the vehicle, and takes values of 0.4, 0.5, or 0.6 m. The calculation takes place in four steps, each step taking into account one of the four basic directions in the pixel matrix (south–north, east–west, southeast–northwest, northeast–southwest). This procedure is necessary with regard to the possibility of moving the vehicle in different directions.

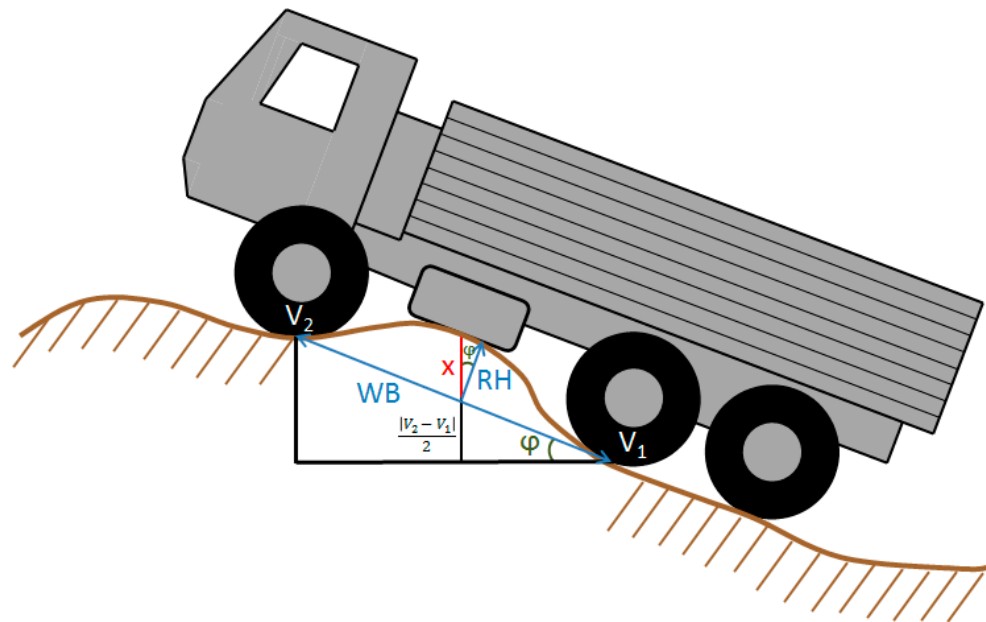

**Figure 4.** The principle of calculation of the altitude of the chassis at the hypothetical point of contact with the terrain between the axles.

The result of the calculation is four rasters, from which the differences between the model chassis altitudes and terrain altitudes when moving in the given direction are calculated. Negative pixel values represent the jam of the vehicle. To improve the terrain edge classification and to optimize the model, the results are sampled to a pixel size of 0.5 m. The resulting raster of the impassable edges is obtained from all four layers and contains a minimum value at each location.

The accuracy of edge identification is again influenced by the accuracy of the DMR5 model and the pixel size of the raster model. The reliability of the calculation was verified again in the terrain using vehicles. Based on the verification, an average height deviation of 0.1 m was calculated. This deviation was considered in the case of classification of impassable terrain edges in a similar way as in the case of field levels.

The procedure for detecting terrain edges is time-consuming and not suitable for large areas (over 1 km$^2$). Therefore, the research team's effort was to find the optimal way to look for terrain edges, taking into account the parameters of a particular vehicle. For this solution, the ProfileCurvature tool was used to calculate the values of the second derivative of the modeled relief pattern. This tool is for curving convex or concave slope curvature (Figure 5):

- A: a negative value indicates that the surface is upwardly convex at that cell;
- B: a positive profile indicates that the surface is upwardly concave at that cell;
- C: a value of zero indicates that the surface is linear

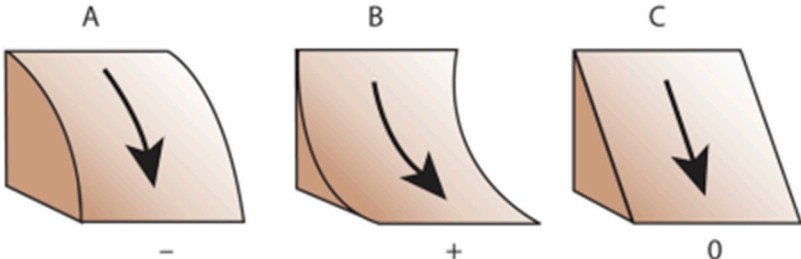

**Figure 5.** The principle of calculating profile curvature. A negative value (**A**) indicates that the surface is upwardly convex at that cell. A positive profile (**B**) indicates that the surface is upwardly concave at that cell. A value of zero indicates that the surface is linear (**C**). [23].

The calculated values in each pixel were compared with the spots of terrain edges detected by the original procedure for each vehicle type. By comparison, one value was found for each vehicle defining the limit value for passability of the terrain edge (*LCur*). In total, seven limit values were set for the basic types of chassis used in the Czech Army. They are Land Rover Defender 110, UAZ 469, Iveco M65E, Pandur II, Tatra T810, T815 6 × 6, and T815 8 × 8. Based on these limit values, terrain edges for these vehicles can be detected very quickly in large areas.

If it is necessary to classify terrain edges for other types of vehicles, it is necessary to perform the calculation and the whole comparison procedure from the start. For this reason, an attempt has been made to find a functional dependence between the limit value of curvature and the technical parameters of the vehicle on which the ability to overcome the terrain edge is dependent (round clearance, wheelbase). Using the values for the above-mentioned vehicles, a search for a suitable interpolation function was performed to determine the dependence between the curvature value and the vehicle parameters. There were several variants, including:

- dependence on wheelbase;
- dependence on the vehicle's ground clearance;
- dependence on multiple wheelbase and light altitude;
- dependence on wheelbase/lightness ratio;
- dependence on the angle defined by the wheelbase and the ground clearance

The latter variant (the dependence of the curvature value on the angle defined by the wheelbase and the ground clearance) proved to be the most appropriate when using the second-degree polynomial (Figure 6). The correctness of this feature confirms that the curvature value is nearing zero with a decreasing height of the vehicle. Reliability of the selected regression function is 0.992. The initial procedure was to calculate the curvature limit value for Land Rover Defender 130 and Tatra T815 4 × 4. The values obtained were compared with the values calculated by this function and in both cases there is a minimum difference between the values.

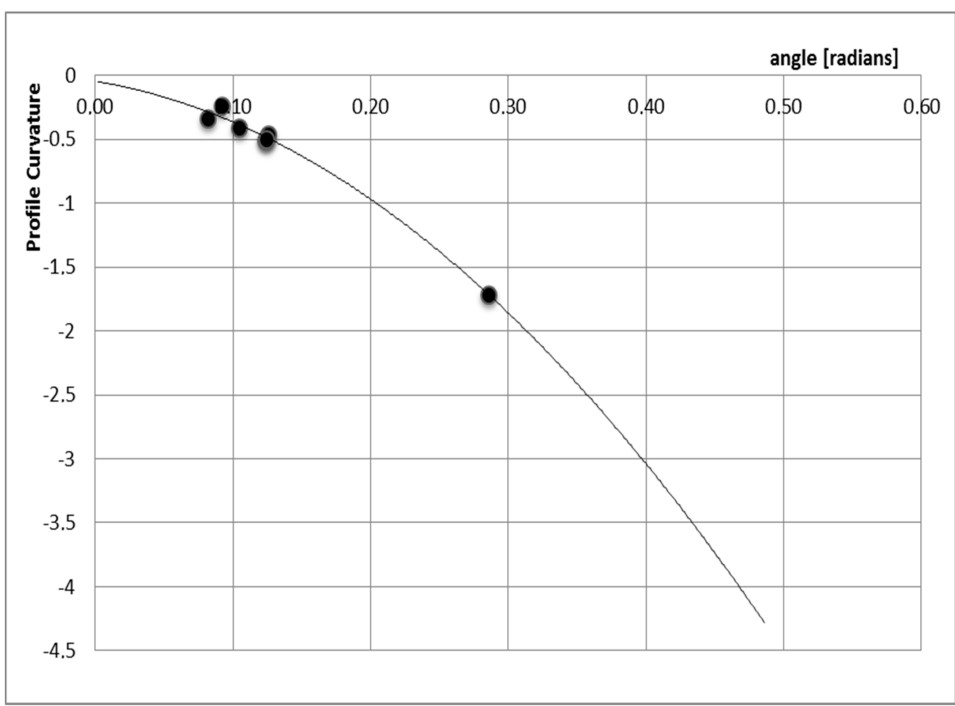

**Figure 6.** The dependence of the curvature value on the angle defined by the wheelbase and the ground clearance.

The classification of terrain edge passability is divided into the following three groups due to possible inaccuracies in the embossed model and vertical movements of the vehicle (dampers, tires, etc.):

- impassable edges of the relief (*Curvature* ≤ *LCur* + *0.1LCur*);
- probably impassable relief edges (*Curvature* > *LCur* + *0.1LCur* ∧ *Curvature* < *LCur* − *0.1LCur*);
- passable edges of the relief (*Curvature* > *LCur* − *0.1LCur*)

### 3.3. Identification of Impassable Angles of Approach

The proposed procedure for identifying impassable approach angles works with three technical parameters of the vehicle. The deformation of springs, tires, and terrain deformations due to vehicle load is not considered in the calculation. The used technical parameters are:

- forward approach angle;
- the front length of the vehicle (distance between the center of the first axle and the front of the chassis);
- wheelbase of the front and rear axles

The entire computational algorithm is based on map algebra and works with the approach slope value, the approach angle, and the calculated slope value of the vehicle relative to the chosen location (Figure 7). The size of the pixel is chosen with respect to the dimensions of the vehicle as 0.4, 0.5, or 0.6 m. The specific value is selected depending on the wheelbase of vehicle so that the number of pixels is odd. The odd number of pixels is important for the calculation process. A difference ±0.1 meter from a standard pixel size of 0.5 meters makes the location of the axles on the ground more precise to the actual microrelief object, thus eliminating possible calculation errors. In addition to the aforementioned values, the calculation in this model also takes into account the slope orientation, which was not necessary in the previous cases. In each direction, the slope and slope orientation are analyzed. The calculation of the modified angle of approach (*MAoA*) for each pixel follows, depending on the longitudinal tilt of the vehicle in the given direction. This calculated value is then

compared to the angle of relief (*AoR*) at the given location. With regards to the possibility of moving the vehicle in different directions, the calculation takes place in four steps, each step taking into account two of the eight directions in the pixel matrix (south–north and opposite, east–west and opposite, southeast–northwest and opposite, northeast–southwest and opposite).

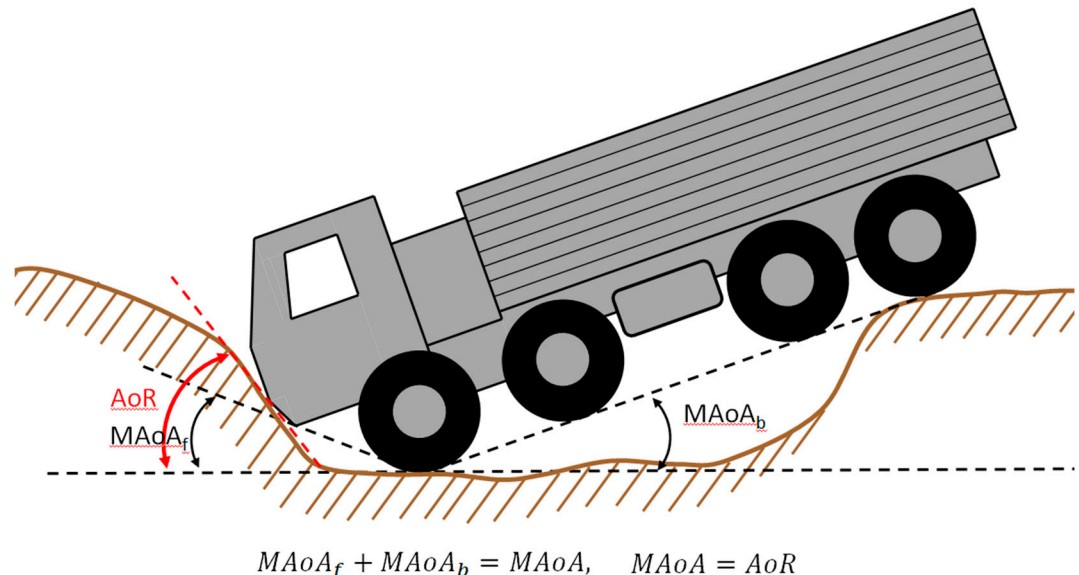

$$MAoA_f + MAoA_b = MAoA, \quad MAoA = AoR$$

**Figure 7.** The principle of the calculation of the modified angle of approach (*MAoA*).

As a part of the classification of the approach angles, the calculated values are again divided into the following three groups:

- impassable angles of approach ($AoR \geq MAoAr + 0.1MAoA$);
- probably impassable angles of approach ($AoR < MAoA + 0.1MAoA \wedge AoR > MAoA - 0.1MAoA$);
- passable angles of approach ($AoR < MAoA - 0.1MAoA$)

The entire calculation is very time-consuming and it has not been possible to resolve the optimization of the calculation. It is clear from the field tests that the occurrence of these impassable approach angles is not negligible, and despite the complexity of the calculation it is necessary to be performed.

### 3.4. Example of Occurrence of Impassable Microrelief Objects

The importance of searching of impassable microrelief objects is documented in the following examples. Two different areas were selected for illustration. Area number one represents a flat territory in the Dolnomoravsky uval near the town of Rakvice (Figure 8a,c). Area number two features a rugged relief and is located near Nedvedice in the Hornosvatecka vrchovina (Figure 8b,d). The size of each area is 1 × 1 km. Identification of impassable microrelief objects was performed in both areas for vehicles Land Rover Defender 110 (LR110) and Tatra T815 8 × 8 (T815). The results of the calculations are shown in Figures 9 and 10. The impassable microrelief objects are displayed in the following colors:

- red—impassable perpendicular stages;
- black—impassable angles of approach;
- blue—impassable sharp edges

The results show that the movement of vehicles in the field is limited by the amount and shape of the microrelief objects in area. Larger quantities of impassable microrelief objects have been identified in a rugged relief. Impassable microrelief objects also occur in a flat relief, especially in the vicinity of watercourses and main roads.

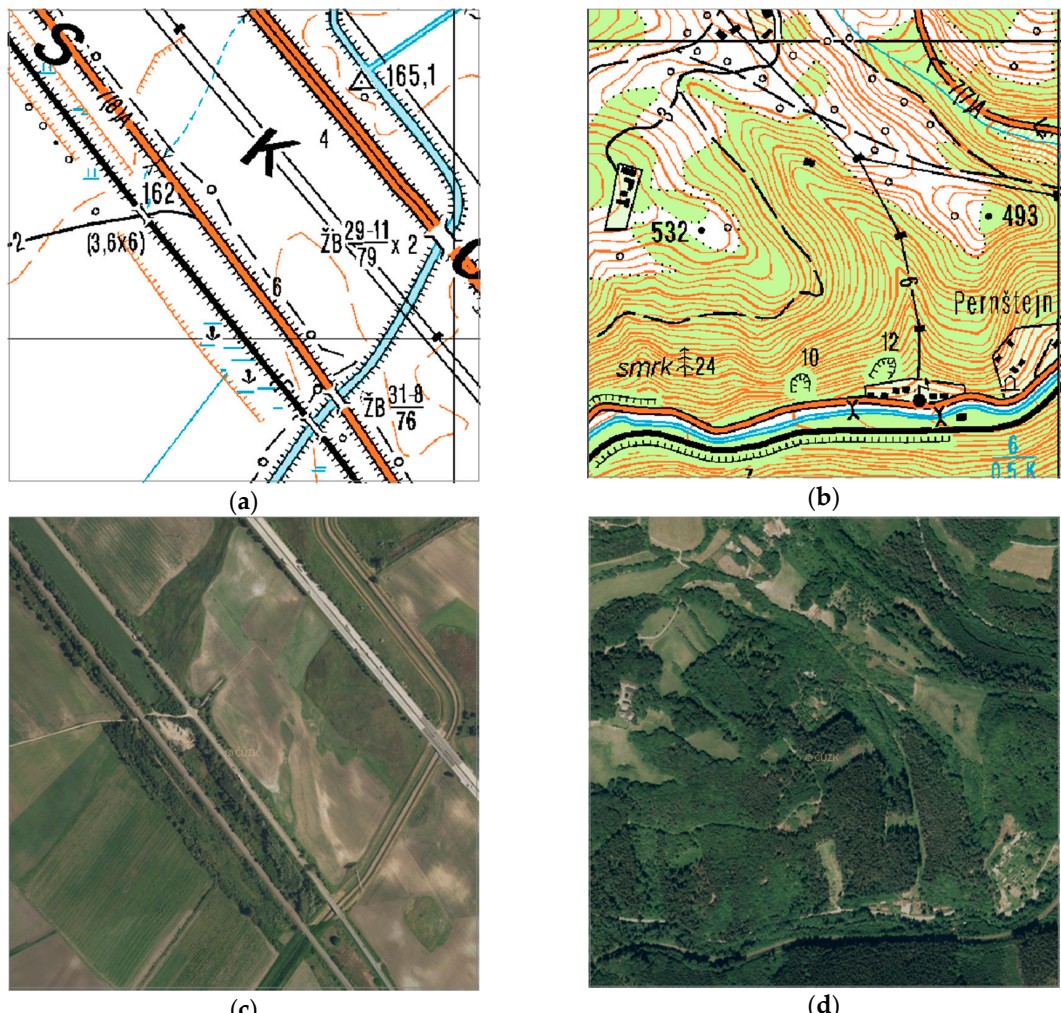

**Figure 8.** Testing areas: (**a**) Rakvice location on a topographic map; (**b**) Nedvedice location on a topographic map; (**c**) Rakvice location on a ortophotomap; (**d**) Nedvedice location on a orthophotomap. Topographic map © Ministry of Defence Czech Republic. Orthophotomap © Ministry of Defence Czech Republic and State Administration of Land Surveying and Cadastre.

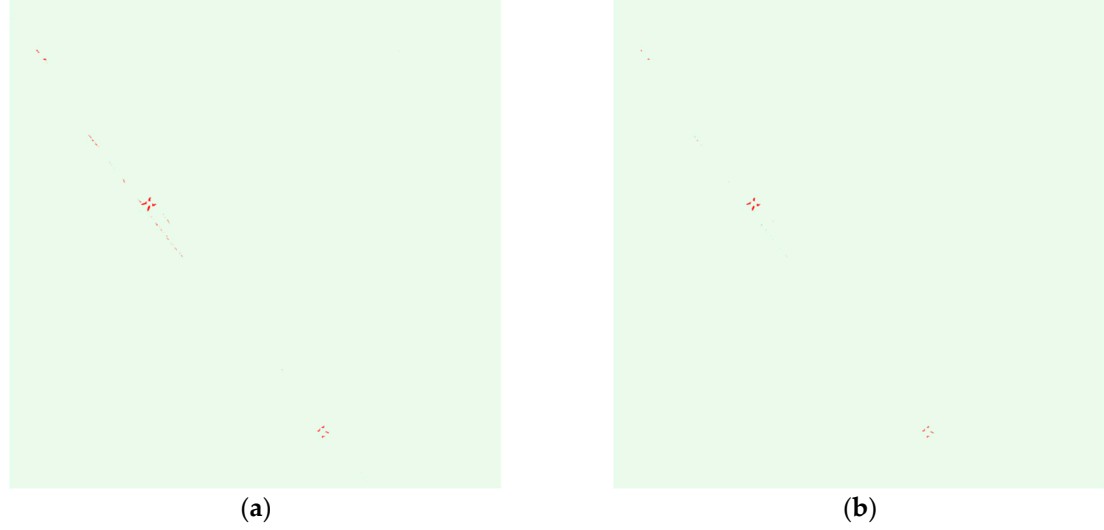

**Figure 9.** *Cont.*

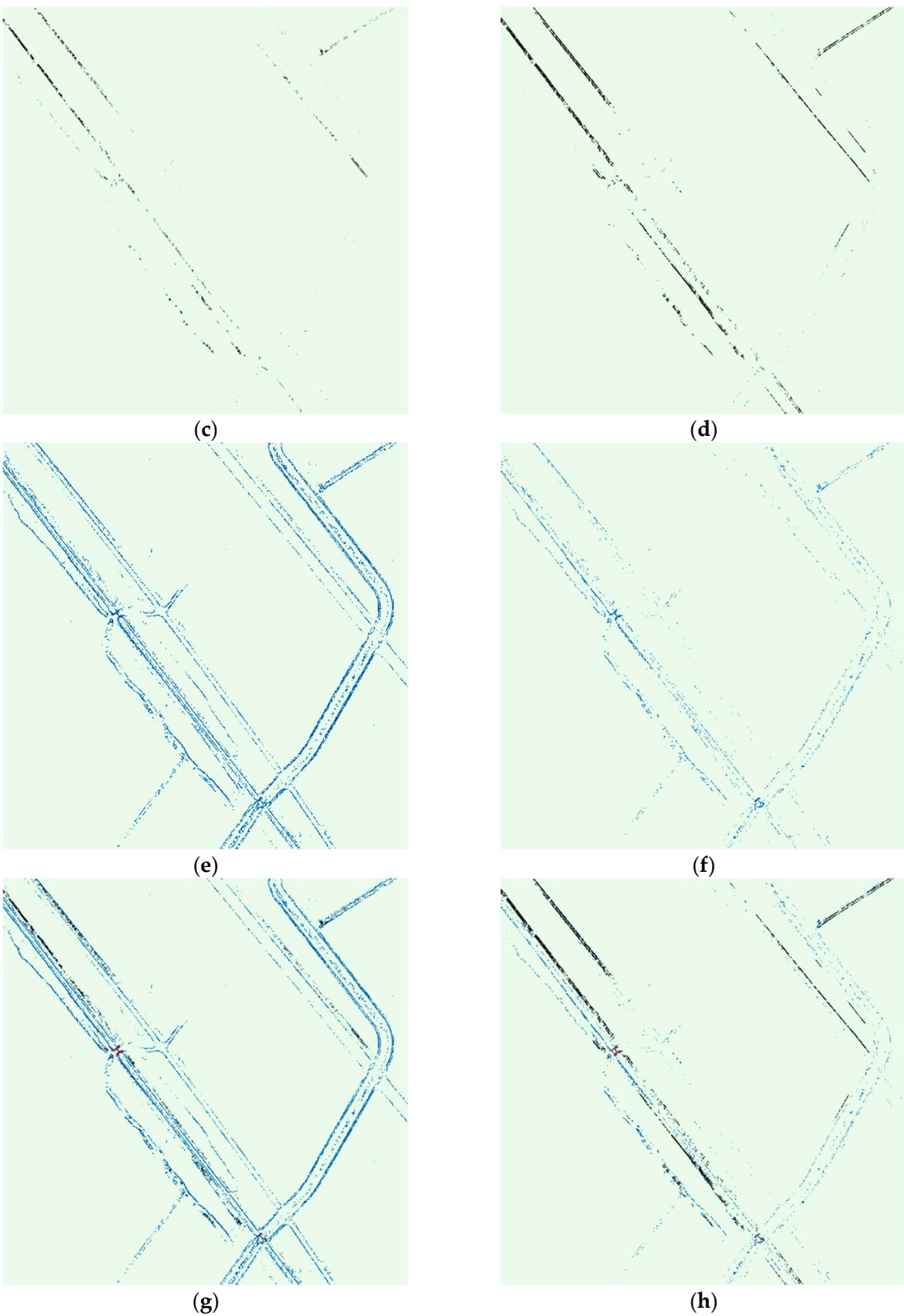

**Figure 9.** Microrelief objects detected in area Rakvice: (**a**) Impassable perpendicular stages for LR110; (**b**) Impassable perpendicular stages for T815; (**c**) Impassable angles of approach for LR110; (**d**) Impassable angles of approach for T815; (**e**) Impassable sharp edges for LR110; (**f**) Impassable sharp edges for T815; (**g**) All impassable microrelief objects for LR110; (**h**) All impassable microrelief objects for T815.

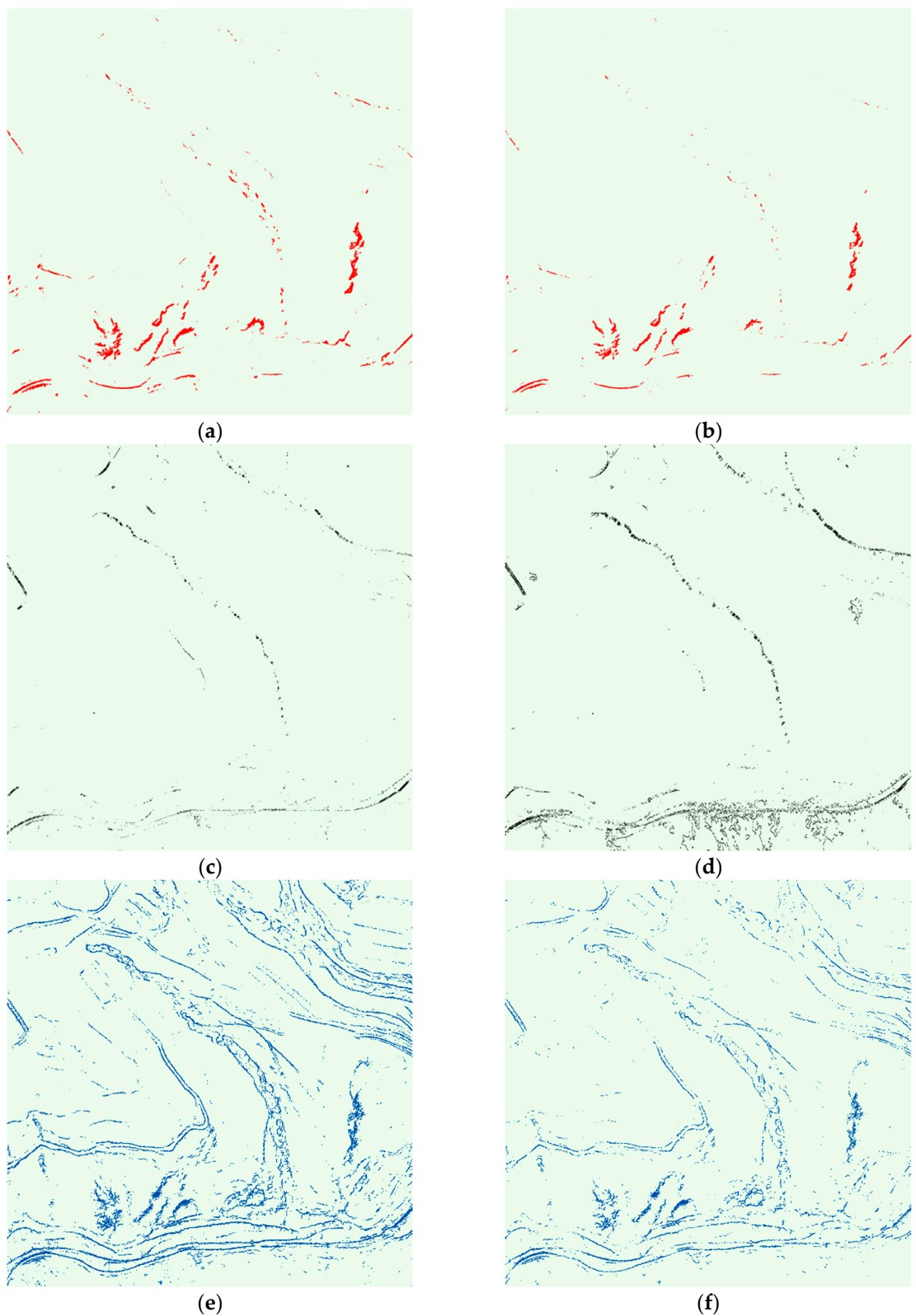

**Figure 10.** *Cont.*

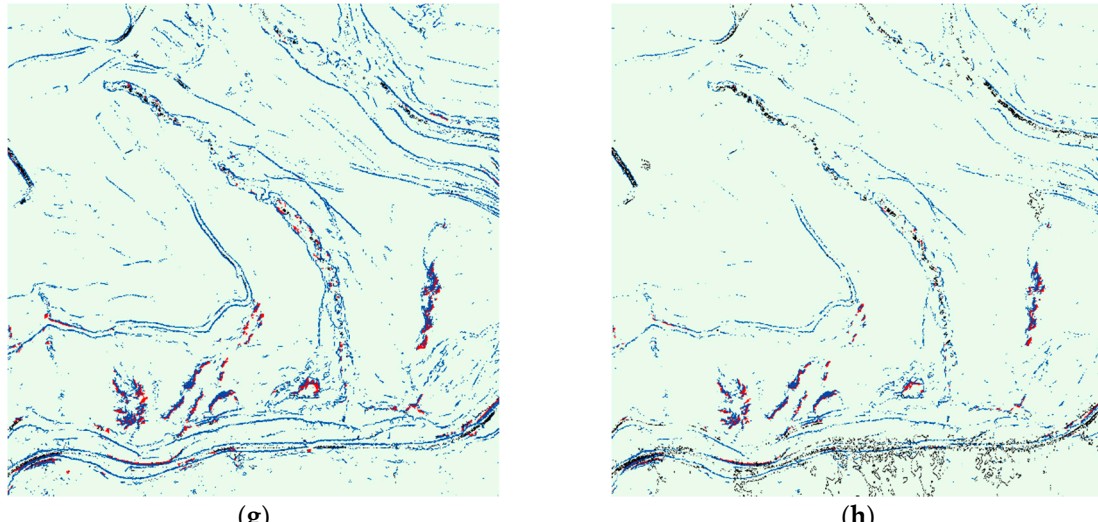

(**g**)  (**h**)

**Figure 10.** Microrelief objects detected in area Nedvedice: (**a**) Impassable perpendicular stages for LR110; (**b**) Impassable perpendicular stages for T815; (**c**) Impassable angles of approach for LR110; (**d**) Impassable angles of approach for T815; (**e**) Impassable sharp edges for LR110; (**f**) Impassable sharp edges for T815; (**g**) All impassable microrelief objects for LR110; (**h**) All impassable microrelief objects for T815.

## 4. Discussion

When evaluating the terrain, the landscape is assessed in its complexity. The evaluation method is dependent on geographic data available, the time options and the target group of the users of the given evaluation. Output of terrain survey for the operational command level will be different from tactical command level output. The main elements, such as communications, vegetation, water, settlements and relief, will always be included. For the tactical stage, other influences such as soil loadability depending on the weather, or the occurrence and overcoming of microrelief objects are also important.

Knowledge of the location of microrelief objects in space and the possibility of overcoming them by military vehicles can have a significant influence on the decision-making of the commanders. Microrelief objects can be used to build barriers to defensive positions in order to direct the enemy's movement to locations suitable for focused fire and technology destruction, as well as for hiding people and vehicles. Therefore, it is important to know their size, shape, and method of spatial deployment.

A new generation of relief models, created from airborne laser scanning data, can analyze the elements of a microrelief and thereby provide the commanders with the information they need to make decisions. Detecting obstacles using these models is a highly promising way to evaluate the character of terrain. The previous text presents a model design for identification of impassable microrelief objects utilizing vehicle technical parameters, map algebra tools, and built-in ArcGIS functions.

The development process of this model was gradually based on individual cases that may occur when overcoming microrelief objects. The accuracy of the modeling itself is influenced by the overall precision of the used elevation model, the DMR5 inaccuracies in the locations of the microrelief shapes and the pixel sizes of the raster model entering the calculation. As part of the development, the model was gradually modified to its present form and the results of field calculations were continually tested.

The verification of terrain modeling results was organizationally challenging. In addition to finding suitable test sites, it was necessary to secure the tested vehicles, as well as recovery vehicles and health care. Because of the deployment of military technology, all tested sites were bound to military training areas. Two tests were conducted. The first one took place in May 2017 in Military Training Area (MTA) Libava and the other in March 2018 in the Military Training Area Hradiste.

Several microrelief objects were selected for verification, which were gradually overtaken by various types of vehicles (Figure 11). Based on the initial verification in MTA Libavá, partial

inaccuracies in edge detection were identified. They were mainly based on the evaluation of the passable edges as impassable. This is to a certain extent linked to drivers' driving abilities, driver knowledge of the location, the character of the relief before the actual microrelief object, and the inaccuracies caused by the interpolation of point data into a continuous height model. On the basis of these facts, the model was modified and the identification results were divided into three categories, one of which expresses the uncertainty in the possibility of overcoming the microrelief object. Reliability of detection of impassable microrelief objects using DMR5 achieved 75–90% depending on the type of obstacle compared to field testing. A total of 38 rides were carried out on 5 microrelief obstacles.

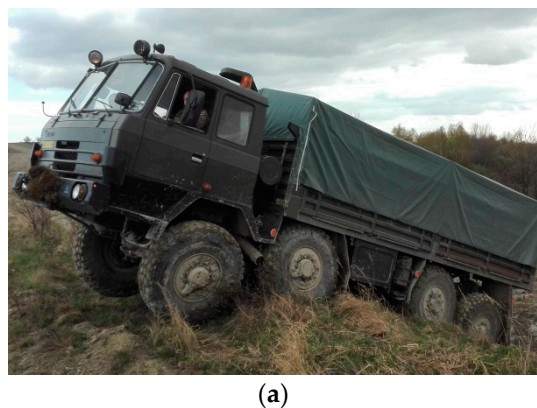 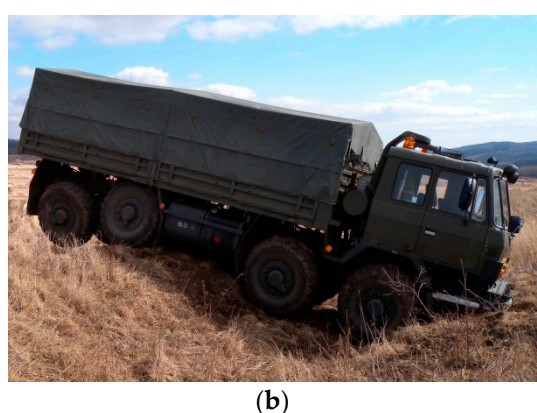

(**a**)                                             (**b**)

**Figure 11.** Testing the reliability of the identification of microrelief objects in the terrain: (**a**) T815 overcame sharp edge; (**b**) T815 was stuck due to its small approach angle.

In addition to verifying the results of vehicle modeling, all objects were targeted by an electronic tachymeter and their shape was compared to the shape obtained from the DMR5 data. The comparison confirmed the model's tendency to smooth out the terrain edges. When comparing terrain-based profiles and profiles detected from the raster model, the deviation in the critical value on the measured profiles ranges from 10 to 15%. The magnitude of deviations depends on the shape of the microrelief object and its size, larger deviations occur in case of perpendicular edges and smaller objects. From these values, the degree of uncertainty in overcoming individual tasks of overcoming the microrelief object was determined. Currently, a more detailed specification of the embossed model cannot be expected. Therefore, any refinement of the calculation will only be linked to the possible implementation of further tests using military vehicles.

## 5. Conclusions

A few years ago, it was impossible to have accurate information on all microrelief objects from large areas. Users typically only had maps or vector layers of microrelief objects where the most significant objects were put and their type was expressed by a map mark or attribute. The height of the object was additional information for the most important ones. The accurate assessment of the ability to overcome these objects by the vehicle was rather complicated and depended on personal experience of the assessor (military geographer). The assessment was made generally only on the basis of the height of the object without an evaluation of the technical parameters of the individual vehicles. Therefore, it was often subjective and, as a rule, did not fully coincide with the reality in the terrain.

The described procedure for evaluating the passability of microrelief objects using the new generation of height models represents a qualitatively new way of working. The entire process is fully automated and utilizes the real value of the vehicle's technical parameters. The evaluation of the passability of microrelief objects is processed for specific types of vehicles and it is possible to take measures to deploy different types of technology within the planning process. In addition, it eliminates the subjectivity of the passability assessment. The obtained layer of impassable microrelief objects is not intended to find optimal routes. In the future, there is a plan to solve the last case (overcoming

the deep notch) and to gradually implement the whole system of evaluation of the passability of the microrelief objects into the complex model of the passability assessment. The beta version will be launched in in 2019 within the internal army network and made available to the Army of the Czech Republic. This complex model will allow to create a map of passability but also to find optimal routes in the specified area.

The created model is a significant contribution in assessing the passability of the terrain and, in addition to the army, it is also usable for rescue vehicles when moving in the terrain. In addition to its own edge detection, there is a significant finding from field trials, such as the influence of a driver's ability to overcome the microrelief object. Drivers with less experience were not able to overcome some microrelief objects in many cases, even though the technical characteristics of the vehicle allowed it. For inexperienced drivers, it is more difficult to move around in the terrain, and it is necessary to conduct their regular driving training even in the field of overcoming microrelief objects.

**Author Contributions:** Conceptualization and methodology, Filip Dohnal and Martin Hubacek; Model programming, Filip Dohnal; Validation, Filip Dohnal, Martin Hubacek, and Katerina Simkova; Model optimization, Filip Dohnal and Martin Hubacek; Writing the paper Martin Hubacek, Katerina Simkova, and Filip Dohnal.

**Funding:** This paper is a particular result of the defense research project DZRO K-210 NATURENVIR managed by the University of Defense in Brno.

**Acknowledgments:** The authors especially want to thank the members of the Seventh Mechanized Brigade of the Army of the Czech Republic for their professional work during the field tests of passability of microrelief objects with military vehicles.

**Conflicts of Interest:** The authors declare no conflict of interest. The funders had no role in the design of the study; in the collection, analyses, or interpretation of data; in the writing of the manuscript, or in the decision to publish the results.

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
