# Peer review of "Detection of Microrelief Objects to Impede the Movement of Vehicles in Terrain"

_ijgi, doi:10.3390/ijgi8030101_

Round 1
Reviewer 1 Report
The manuscript discusses the terrain analysis for the movement of Military Vehicles. The current title might not reflect the work as the research does not study the “influence” of terrain relief on the movement of (Military) vehicles.
Terrain Analysis using different data including LiDAR data is a general task in GIS community. The contribution of the manuscript is not clear. There are so many well documented studies on terrain analysis as well as suitability analysis to model optimal routes for vehicles movement which should be reviewed and included in the introduction section, then the contribution of the paper should be explained.
The technical parameters (lines 12-13 and 42-43) need some explanation. What are these parameters?
Line 44: “However, the use of digital relief models covering large areas is problematics” is not accurate. During last two decades, so many high resolution/accuracy DEMs, DTMs and topography information for large areas have been created.
Line 202 -203: How is the pixel size selected? What is the relationship between the pixel size and the dimensions of the vehicle? I believe that the data accuracy and analysis types should be taken into account when selecting the pixel size.
How do you validate the research findings?
I’d suggest to do a suitability analysis to identify the optimal route for military vehicles based on your analysis.
Author Response
Thank you for the factual comments on the text.

Reviewer 2 Report
The paper is well structured, the motivation and the state of the art are clearly expressed in Introduction. The results are supported by several examples.
References 4, 7 and 17 are not cited properly. They need to be corrected.
Instead of:
Hofmann, A.; Mayerova, S.; Talhofer, V.; Kovarik, V. Creation of models for calculation of coefficients of terrain passability. Quality & Quantity, 2015, vol. 49, no. 4, p. 1679-1691.
Talhofer, V.; Mayerová, S.; Hofmann, A. Towards efficient use of resources in military: methods for evaluation routes in open terrain. Journal of Security Sustainability Issues, 2016, vol. 6, no. 1, p. 53-70.
Dohnal, F.; Hubacek, M.; Sturcová, M.; Bures, M.; Simkova, K. Identification of Microrelief Shapes Along the Line Objects Over DEM Data and Assessing Their Impact on the Vehicle Movement. In Proceedings of the 2017 International Conference on Military Technologies (ICMT), Brno, Czech Republic, 31 May–2 June 377 2017, p. 262-267.
Based on these comments, I recommend the paper for publication, after the authors will submit a new draft taking into account my remarks and suggestions.
As I have already written in my report, the paper titled „Influence of micro-relief objects on the movement of vehicles in terrain“ is very interesting, it gives new results how the relief of terrain affects the possibilities of vehicles movement off the road. Moreover, the authors are precise in describing the problem and its solution process. They build the model and also successfully verified it in the terrain.
The only thing that would be worth considering to add is in part „Identification of impassable sharp edges“. Here the authors wrote:
The latter variant (the dependence of the curvature value on the angle defined by the wheelbase and the ground clearance) proved to be most appropriate when using the second degree polynomial (Figure 1). The correctness of this feature confirms that the curvature value is limited to zero with a decreasing height of the vehicle.
The procedure is illustrated in the figure, but there is not presented the equation of the function. It is only mentioned that the second degree polynomial curve was used. Also, the indication of its reliability is missing. I would asked the author to add mainly the information concerning the reliability.

Author Response

(The authors gave the same response as above.)

Reviewer 3 Report
In the summary there is no clearly defined:
scientific purpose,
research methods,
partial results and
final results.
Similarly in the content of the article, the scientific goal is blurred. The summary is too general.
In the conclusions, indicate the advantages and disadvantages of the proposed solution (from the point of view of cartography).
No comparison results.
Author Response

(The authors gave the same response as above.)

Round 2
Reviewer 1 Report
The authors revised the paper and provided clarification of the paper contribution.
Author Response
English language and style were check by English teacher and native speaker. Minor modifications were made to the text.
Reviewer 3 Report
The article is ready for publication.
Author Response

(The authors gave the same response as above.)
